# Antiplatelet Therapy Improves the Prognosis of Patients with Hepatocellular Carcinoma

**DOI:** 10.3390/cancers12113215

**Published:** 2020-10-31

**Authors:** Tsuguru Hayashi, Michihiko Shibata, Shinji Oe, Koichiro Miyagawa, Yuichi Honma, Masaru Harada

**Affiliations:** Third Department of Internal Medicine, School of Medicine, University of Occupational and Environmental Health, 1-1 Iseigaoka, Yahatanishi-ku, Kitakyushu 807-8555, Japan; 3shibata@med.uoeh-u.ac.jp (M.S.); ooes@med.uoeh-u.ac.jp (S.O.); koichiro@med.uoeh-u.ac.jp (K.M.); y-homma@med.uoeh-u.ac.jp (Y.H.); msrharada@med.uoeh-u.ac.jp (M.H.)

**Keywords:** hepatocellular carcinoma, antiplatelet therapy, prognosis

## Abstract

**Simple Summary:**

Antiplatelet therapy shows antitumor effect in several types of cancers. In liver disease, antiplatelet therapy reduces liver fibrosis and occurrence of HCC. However, the effect after diagnosis of hepatocellular carcinoma is still unknown. Therefore, we aimed to clarify the effects of antiplatelet therapy for HCC patients in this study. We compared patients with and without antiplatelet therapy in overall survival, liver-related death, tumor progression, Child-Pugh deterioration and bleeding. Our results suggested that antiplatelet therapy had antitumor effect, liver protective effect and safety. Therefore, patients with antiplatelet therapy could reduce liver-related death and improve overall survival.

**Abstract:**

Aims: Antiplatelet therapy has been reported to reduce liver fibrosis and hepatocellular carcinoma (HCC), and has exhibited antitumor properties in other cancers. However, the effects of antiplatelet therapy after diagnosis of HCC are unknown. We investigated the effects of antiplatelet therapy on prognosis, tumor progression, liver function and safety in HCC patients. Methods: We retrospectively analyzed 772 HCC patients. Antiplatelet therapy was defined as the regular intake of aspirin or clopidogrel from HCC diagnosis through to an endpoint of either overall survival (OS) or liver-related death. Overall survival, liver-related death, tumor progression, Child–Pugh deterioration and hemorrhage were analyzed for patients who either had or had not undertaken antiplatelet therapy. Results: The numbers of patients who did and did not undertake antiplatelet therapy were 111 and 661, respectively. Patients who undertook antiplatelet therapy were older and had better liver function at diagnosis. Antiplatelet therapy resulted in significant improvements in OS (*p* < 0.01) and lower risk of liver-related death (*p* < 0.01). Multivariate Cox regression analysis revealed that antiplatelet therapy had a significant negative association with liver-related death (hazard ratio (HR): 0.64, 95% confidence interval (CI): 0.44–0.93, *p* = 0.02). In patients who underwent transcatheter arterial chemoembolization (TACE) as the first treatment, antiplatelet therapy prevented tumor progression (*p* < 0.01) and Child–Pugh deterioration (*p* < 0.01). Antiplatelet therapy did not increase the risk of hemorrhagic events. Conclusions: Antiplatelet therapy reduced liver-related death and improved OS safely in HCC patients.

## 1. Introduction

Hepatocellular carcinoma (HCC) is one of the most common causes of cancer-related death worldwide [1]. Surgical resection, radiofrequency ablation (RFA), transcatheter arterial chemoembolization (TACE) and tyrosine kinase inhibitor (TKI) have become the standard options for HCC treatment. However, prognosis in HCC remains unsatisfactory because of a high recurrence rate and deterioration of liver function after treatment [2].

In previous studies, antiplatelet therapy had been shown to prevent liver fibrosis and the occurrence of HCC [3,4,5,6,7]. Several mechanisms have been proposed to explain the protective effects. First, the anti-inflammatory action of antiplatelet agents inhibits transcription of nuclear factor-kappa B (NF-κB) in the endothelial cells and prevents adhesion of macrophages and T-lymphocytes [8]. It also reduces the level of inflammatory cytokines including tumor necrosis factor-β (TNF-β) and platelet derived growth factor (PDGF). Second, antiplatelet agents have antithrombotic effects which prevent liver fibrosis [9]. Aspirin and clopidogrel were compared in the occurrence of HCC. Both drugs reduced the risk of HCC. However, hazard ratio of HCC development was lower in aspirin in time varying cox proportional hazards regression analysis [5].

Furthermore, antiplatelet agents have antitumor effects. Aspirin inhibits cyclooxygenase (COX)-2, which promotes inflammation and cell proliferation [10]. In addition, other mechanisms such as the inhibition of NF-κB, induction of apoptosis and catabolism of polyamines have been investigated [11,12]. Aspirin has been shown to decrease the risk of recurrence and death in patients with colorectal cancer, breast cancer and lung cancer [13,14]. Clopidogrel has a different mechanism from aspirin, in that it inhibits the P2Y12 adenosine diphosphate receptor [15,16]. Inhibition of platelet activity via the P2Y12-dependent mechanism reduced the development of tumor metastasis [17]. In fact, clopidogrel reduced the risk of metastasis of colon cancer and prostate cancer [18,19].

However, the effect of antiplatelet therapy after diagnosis of HCC is unknown. The recent epidemics of obesity and metabolic syndrome are associated with an increasing prevalence of nonalcoholic fatty liver disease (NAFLD) [20,21], which is a major component of non-viral HCC [22]. Underlying conditions such as hyperlipidemia, diabetes mellitus and hypertension increase the risk of coronary artery disease and cerebrovascular disease. Antiplatelet therapy is useful for prevention of these arterial diseases. The number of HCC patients with antiplatelet therapy is currently increasing and is expected to increase further in the future. Thus, there is a need to clarify the effects and safety of antiplatelet therapy for HCC treatment.

In this study, we investigated the effect of antiplatelet therapy on the prognoses of HCC patients, focusing particularly on HCC recurrence and deterioration of liver function. We also evaluated the safety of antiplatelet therapy by calculating the incidence of hemorrhagic events.

## 2. Results

### 2.1. Patient Characteristics

We evaluated the eligibility of 784 patients with HCC, and 772 were subsequently included in this study. Twelve patients were excluded because they started antiplatelet therapy after the first HCC treatment. Among the 772 patients, 111 (14.4%) received antiplatelet therapy from the time of diagnosis of HCC (Appendix A). The numbers of patients who took aspirin, clopidogrel and both were 96 (86.5%), 18 (16.2%) and 3 (2.7%), respectively. Antiplatelet therapy was administered for coronary artery diseases (*n* = 46, 41.4%), cerebrovascular disease (*n* = 38, 34.2%) and peripheral arterial disease (*n* = 3, 2.7%). For 24 patients, the reason for administration was not clear. The baseline characteristics of HCC patients who did or did not undergo antiplatelet therapy are shown in Table 1. Patients undertaking antiplatelet therapy were older, and the percentage of males in this group was significantly higher. The liver function undertaking with antiplatelet therapy was better than those who did not undertake antiplatelet therapy. The proportions of HCC staging and alpha-fetoprotein (AFP) levels differed significantly. The first treatment methods did not differ between the two groups.

### 2.2. Prognosis of HCC Patients

The median observational period after the first treatment was 1.9 years (maximum 13.6 years). The median overall survival (OS) was 3.5 years in all patients. Median OS in patients who did or did not undertake antiplatelet therapy were 5.9 years and 3.1 years, respectively. In addition, 5-year survival in patients who undertook antiplatelet therapy was significantly better than for those who did not (59.0% versus 33.5%, *p* < 0.001) (Figure 1a). There was a significant difference between the numbers of liver-related deaths in patients who did or did not undertake antiplatelet therapy (five-year rate, 36.1% versus 62.4%, *p* < 0.001) (Figure 1b). Univariate Cox regression analysis of liver-related deaths found that the Child–Pugh score, HCC stage, AFP, des-gamma-carboxy prothrombin (DCP), first treatment method and antiplatelet therapy were significant factors (Table 2). Multivariate analysis showed that antiplatelet therapy was a significant negative factor for liver-related death (hazard ratio (HR): 0.640, 95% confidence interval (CI): 0.445–0.926, *p* = 0.002), along with Child–Pugh score, HCC stage, tumor markers and first treatment method (Table 2). The cut-off value for tumor markers was defined according to previous studies. The causes of death in patients who undertook antiplatelet therapy were liver-related death (92.8%), acute pneumonia (2.4%), malignant lymphoma (2.4%) and unknown factors (2.4%). When we investigated OS according to the etiology, the significant effect of antiplatelet therapy was shown in non-hepatitis B virus (HBV) and non-hepatitis C virus (HCV) (NBNC) (5-year rate, 45.7% versus 31.7%, *p* = 0.003). However, in HBV (five-year rate, 52.4% versus 34.6%, *p* = 0.077) and HCV (five-year rate, 64.5% versus 34.5%, *p* = 0.078), the effect was not significant.

### 2.3. Propensity Score Matching

The baseline characteristics were different in the two groups. Therefore, propensity score matching was used to adjust patient demographics. After propensity score matching, patient baseline characteristics were similar in the two groups (Appendix A). Antiplatelet therapy was associated with better prognosis. Five-year survival in patients with antiplatelet therapy was significantly better than that without antiplatelet therapy (59.7% versus 32.9%, *p* = 0.01) (Appendix A).

### 2.4. HCC Recurrence and Time to Progression

In patients who underwent curative therapies, there was no difference in HCC recurrence between patients who did or did not undertake antiplatelet therapy (five-year rate, 69.6% versus 74.9%, *p* = 0.26) (Figure 2a). However, patients who underwent TACE with antiplatelet therapy showed significantly longer times to tumor progression (TTP) than those without antiplatelet therapy (five-year rate, 80.0% versus 97.1%, *p* = 0.005) (Figure 2b). In the multivariate analysis, antiplatelet therapy was a significant independent negative factor for tumor progression in patients who underwent TACE (HR: 0.690, 95% CI: 0.484–0.983, *p* = 0.040), in addition to HCC stage and DCP (Table 3).

### 2.5. Deterioration of Liver Function

The numbers of Child–Pugh grade A liver disease in patients who did and did not undertake antiplatelet therapy were 97 (87.4%) and 450 (68.1%), respectively. The three- and five-year cumulative incidences of decompensation in all patients were 46.8% and 63.7%, respectively. Comparing patients who took antiplatelet therapy to those who did not, we found that time to decompensation was significantly longer in the antiplatelet therapy group (five-year rate, 47.6% versus 66.7%, *p* < 0.001). Antiplatelet therapy was a significant independent factor for maintaining good liver function (Child–Pugh grade A) in multivariate analysis (HR: 0.467, 95% CI: 0.321–0.679, *p* < 0.001) (Table 4). In patients who underwent curative therapy at the first treatment, time to decompensation in patients who undertook antiplatelet therapy was longer than in those without antiplatelet therapy (five-year rate, 30.0% versus 48.6%), but the results was not significant (*p* = 0.08) (Figure 3a). Conversely, in patients undergoing TACE, time to decompensation was significantly longer in the antiplatelet therapy group (five-year rate, 59.6% versus 82.4%, *p* < 0.001) (Figure 3b).

### 2.6. Hemorrhagic Events

Hemorrhagic events occurred in 113 patients (14.6%). They consisted of gastrointestinal bleeding (*n* = 79), HCC rupture (*n* = 16), cerebral bleeding (*n* = 10), subcutaneous bleeding (*n* = 5), bile duct bleeding (*n* = 2) and nasal bleeding (*n* = 1). Cumulative incidences of hemorrhagic events in the antiplatelet therapy group showed no difference from the non-antiplatelet therapy group (five-year rate, 23.1% versus 21.4%, *p* = 0.33) (Figure 4). In the multivariate analysis, significant factors for hemorrhagic events were Child–Pugh score (HR: 1.48, 95% CI: 1.33–1.65, *p* < 0.001), HCC stage III (HR: 2.00, 95% CI: 1.07–3.71, *p* = 0.029), HCC stage IVa (HR: 6.24, 95% CI: 2.73–14.3 *p* < 0.001) and HCC stage IVb (HR: 6.71, 95% CI: 2.67–16.9, *p* < 0.001).

## 3. Discussion

In this study, we demonstrated that antiplatelet therapy was associated with improvement in OS and reduction in liver-related deaths in HCC patients. Patients who could not undergo curative therapy showed particular improvements in OS with antiplatelet therapy. The features of patients who undertook antiplatelet therapy at diagnosis were advanced age, high proportion of males and good liver function. In patients who underwent curative therapies as the first treatment, antiplatelet therapy did not have a preventive effect in respect of HCC recurrence. However, antiplatelet therapy tended to delay the deterioration of liver function. Conversely, in patients undergoing TACE at first treatment, both TTP and time to Child–Pugh deterioration were significantly longer in patients who undertook antiplatelet therapy. In the multivariate analyses, antiplatelet therapy was a significant independent factor for preventing liver-related deaths and decompensation in all patients and for improving TTP in patients undergoing TACE at first treatment. These results indicated that antiplatelet therapy has preventative effects on both tumor progression and deterioration of liver function. Therefore, antiplatelet therapy was considered to decrease liver-related death and improve OS after the diagnosis of HCC.

Antiplatelet therapy has previously been shown to prevent liver fibrosis and HCC development [3,4,5,6,7]. In a nationwide study, Tracey et al. demonstrated that aspirin use was associated not only with incidence of HCC but also with liver-related death [7]. This clinical evidence supports the apparent protective effect of antiplatelet therapy on the liver. This protective effect is consistent with our findings that older patients who received antiplatelet therapy had significantly better liver function at HCC diagnosis. These findings suggest that antiplatelet therapy could help maintain good liver function and delay HCC progression. Although many antiplatelet agents, including aspirin, clopidogrel, ticlopidine, dipyridamole and cilostazol, have been shown to have a protective effect on liver fibrosis in animal models [23,24], only aspirin and clopidogrel have demonstrated a preventative effect on liver fibrosis and reduction of HCC development in clinical studies [4]. A previous study compared the effect of aspirin and clopidogrel in relation to HCC occurrence [5]. In this study, the OS of patients taking aspirin did not differ significantly from that of patients taking clopidogrel. However, the number of patients taking clopidogrel was small. Therefore, a further study to compare the two drugs is needed.

Despite the large number of studies regarding the association between the incidence of HCC and antiplatelet therapy, only a few have investigated the effect of antiplatelet therapy on OS in HCC patients. One such study showed that antiplatelet therapy improved OS in patients who underwent liver resection [25]. Another study showed that aspirin improved the prognoses of HCC patients after TACE, with improvement in liver function [26]. However, these studies had some limitations, such as small samples, short observational periods, short administration periods of antiplatelet agents and limited etiology. We therefore used a large sample size with various etiologies in our study and observed for a maximum 13.6 years, demonstrating both the antitumor and protective effects of antiplatelet therapy on the liver.

It is well known that the prognoses of HCC patients depend on tumor burden and liver function. To improve prognosis, it is important to suppress tumor progression and preserve liver function. However, tumor burden and liver function deterioration are strongly correlated. When tumor downstaging cannot be achieved, liver function deterioration cannot be prevented [2]. Therefore, it is difficult to preserve liver function. When liver function progresses to Child–Pugh grades B or C, other treatments such as TKI are not be available. In addition, other drugs that maintain liver function, such as branched-chain amino acids, are limited. We therefore have to determine the most suitable methods for preserving good liver function.

Our findings showed that among patients who underwent curative therapy, there was no difference in HCC recurrence between patients who dd or did not undertake antiplatelet therapy. However, patients who undertook antiplatelet therapy had a tendency to maintain good liver function for longer periods. These results suggest that antiplatelet therapy has a protective effect on the liver. Conversely, patients who underwent TACE with antiplatelet therapy had longer TTP and time to decompensation than those who did not undertake antiplatelet therapy. This could be explained by the antitumor and protective effects of antiplatelet therapy on the liver. Some kinds of cancer cells lead to activation of antiplatelet. Activated platelet releases transforming growth factor and products epithelial mesenchymal transition, leading to tumor progression. Moreover, platelet activation causes stellate cell activation, angiogenesis and drug resistance. This is also reported in HCC [27]. To make matters worse, platelet aggregation causes tumor metastasis. Antiplatelet therapy could surpass these mechanisms.

COX-2 is strongly expressed in human HCC tissues and promotes inflammation and cell proliferation [28]. Therefore, inhibition of COX-2 could prevent tumor growth and metastasis. In fact, aspirin decreases mortality in colorectal cancer, in which COX-2 is overexpressed [13]. In our study, antiplatelet therapy had an antitumor effect in patients who underwent TACE. However, antiplatelet therapy showed no effect in those who underwent curative therapy. Therefore, it is likely that some mechanisms other than COX-2 inhibition exist. Two mechanisms have been proposed to explain this result. First, aspirin has an anti-angiogenic effect. TACE induces ischemia and promotes hypoxia-induced angiogenesis, which promotes tumor growth [29]. Aspirin inhibits hypoxia-induced angiogenesis. Second, aspirin increases sensitivity to various anticancer agents. It has been reported that a combination of aspirin plus anticancer agents, such as doxorubicin, cisplatin and 5-fluorouracil, enhances antitumor effects in cancer cell lines [30,31,32,33]. Thus, aspirin may improve chemosensitivity in TACE. These mechanisms could support the antitumor effects of antiplatelet therapy after TACE.

Our study had some limitations. First, this was a single-center, retrospective study. A prospective study with a large sample size is needed. Second, we could not exclude the influence of direct-acting antiviral agents on liver functions in patients with HCV. Similarly, the clinical significance of nucleotide analogs in HBV patients was not taken into consideration. Third, other medicines that could have antitumor effects were not taking into consideration in this study. However, the number of patients who use these medicines, such as metformin, nonsteroidal anti-inflammatory drugs and branched chain amino acids, from diagnosis to death are a few. Fourth, in the antiplatelet therapy group, they received antiplatelet therapy at diagnosis of HCC. However, we do not consider when they started antiplatelet therapy.

In conclusion, antiplatelet therapy preserved liver function and prevented tumor progression. Therefore, antiplatelet therapy could reduce liver-related death rates and improve OS in patients with HCC.

## 4. Materials and Methods

### 4.1. Patients

This retrospective study enrolled consecutive naïve patients with HCC who started treatment at our university hospital from February 2005 to January 2020. HCC diagnosis was based on histology or radiological findings, such as contrast-enhanced computed tomography (CT) or contrast-enhanced magnetic resonance imaging (MRI). Tumor node metastasis (TNM) stage, which was proposed by the Liver Cancer Study Group of Japan, 6th edition (LCSGJ 6th), was used for evaluation of HCC staging. Underlying liver diseases were categorized according to the presence of chronic hepatitis virus infection. Patients who tested positive for hepatitis B surface antigens and hepatitis C antibodies were classified into hepatitis B virus (HBV) and hepatitis C virus (HCV) groups, respectively. Those who were virus-negative were included in the non-HBV and non-HCV (NBNC) group. HCC treatment was categorized into four groups: curative therapies, TACE, others and best supportive care (BSC). We did not include patients who had undergone liver transplantation. Liver resection and RFA were defined as curative therapy. Other treatments, such as radiotherapy and tyrosine kinase inhibitor (TKI), were defined as “others”. Measurement of blood samples, including alfa-fetoprotein (AFP) and des-gamma-carboxy prothrombin (DCP), was performed before the first treatment in all patients. After the first treatment, patients underwent regular follow-ups with measurement of blood samples and radiologic examinations of CT or MRI every 1 to 3 months. HCC recurrence was diagnosed according to radiological findings and tumor markers. Additional treatments for HCC recurrence were determined according to tumor and liver condition. This study was approved by the institutional ethical board in accordance with the Declaration of Helsinki (H29-078).

### 4.2. Definition of Antiplatelet Therapy

We defined regular medication with aspirin or clopidogrel as antiplatelet therapy. We included patients who had used aspirin or clopidogrel from diagnosis of HCC to the study endpoint. We excluded patients who had started antiplatelet therapy after the diagnosis of HCC.

### 4.3. Endpoint of this Study

The endpoints of this study were overall survival (OS) and liver-related death. OS was the time from the first treatment to death from any cause or last follow-up. We defined all death caused by liver diseases as liver-related death. Liver-related death included gastrointestinal hemorrhage, such as esophageal varices rupture. We also evaluated time to tumor progression (TTP), time to decompensation and time to hemorrhagic events. TTP and time to decompensation were stratified by the first treatment, curative treatment and TACE. Time to decompensation was defined as the time it took for patients with Child–Pugh grade A liver disease at baseline to progress to Child–Pugh grades B or C. Hemorrhagic events included all hemorrhage diseases such as gastrointestinal bleeding, cerebral bleeding and HCC rupture.

### 4.4. Statistical Analysis

All categorical variables were analyzed using the χ2-test or Fisher’s exact test, and continuous variables were compared using Mann–Whitney’s U test. *p*-value < 0.05 was considered statistically significant. We compared age, sex, etiology, liver function, tumor factors at diagnosis of HCC and first treatment method between patients who had and had not taken antiplatelet therapy. OS, liver-related death, TTP, time to decompensation and time to hemorrhagic events were evaluated using the Kaplan–Meier curve, and differences between the two groups were assessed using the log-rank test. A Cox proportional hazards model was used to determine factors associated with endpoints. Continuous numeric variables were expressed as median and interquartile range (IQR). All statistical analyses were performed using the Statistical Package for the Social Science (SPSS) version 25 (SPSS Inc., Chicago, IL, USA) and Easy R (EZR) version 1.29 (Saitama Medical center, Jichi Medical University, Saitama, Japan), and graphical use interface for R (The R Foundation for Statistical Computing, Vienna, Austria) [34].

### 4.5. Propensity Score Matching

Propensity score was estimated using a logistics regression model with the following six variables: age, Child–Pugh score, HCC stage, AFP, DCP and treatment method. We used propensity scores to carry out one-to-one nearest neighbor matching within a caliper of 0.2. Propensity score matching results in the selection of 186 patients (antiplatelet therapy, *n* = 93; no antiplatelet therapy, *n* = 93).

## 5. Conclusions

Antiplatelet therapy preserved liver function and prevented tumor progression after HCC treatment. Therefore, liver-related deaths decreased in HCC patients who undertook antiplatelet therapy.

## Figures and Tables

**Figure 1 cancers-12-03215-f001:**
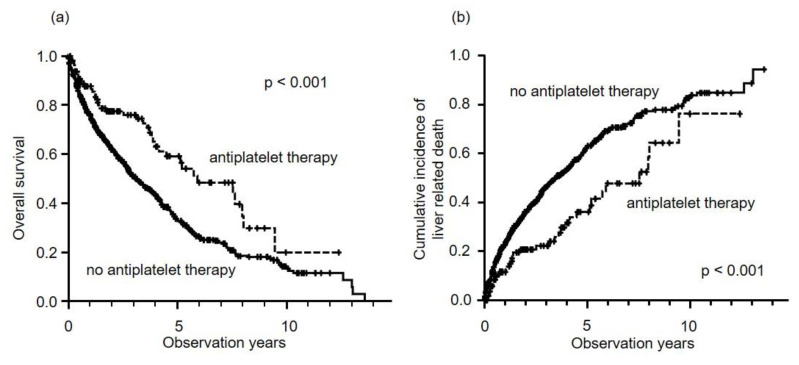
Overall survival and cumulative incidence of liver-related deaths in HCC patients who did or did not undertake antiplatelet therapy. (**a**) Five-year overall survival rates for antiplatelet therapy (dotted line) and no antiplatelet therapy (solid line) were 59.0% and 33.5%, respectively. Antiplatelet therapy resulted in significantly better prognoses (log-rank test; *p* < 0.001). (**b**) Five-year cumulative incidence rates of liver-related deaths with antiplatelet therapy (dotted line) and no antiplatelet therapy (solid line) were 36.1 and 62.4%, respectively. Antiplatelet therapy resulted in significantly better prognoses (log-rank test; *p* < 0.001).

**Figure 2 cancers-12-03215-f002:**
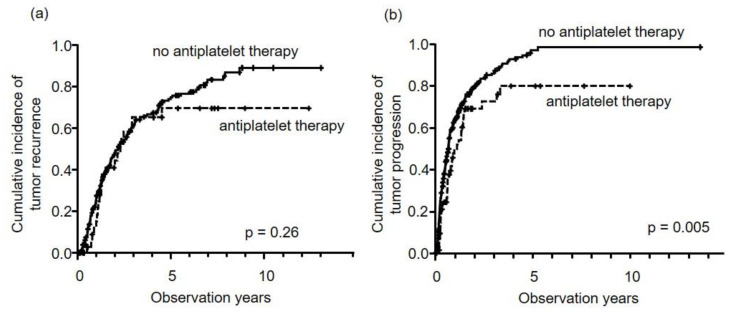
Cumulative incidence of tumor progression stratified by first treatment method, i.e., curative therapy (**a**) and TACE (**b**). In curative therapy (**a**), antiplatelet therapy (dotted line) had no effect on HCC recurrence (*p* = 0.26), compared to no antiplatelet therapy (solid line). However, in TACE (**b**), antiplatelet therapy was associated with significantly longer times to tumor progression. Five-year cumulative incidence rates for antiplatelet therapy (dotted line) and no antiplatelet therapy (solid line) were 80.0% and 97.1%, respectively (*p* = 0.005).

**Figure 3 cancers-12-03215-f003:**
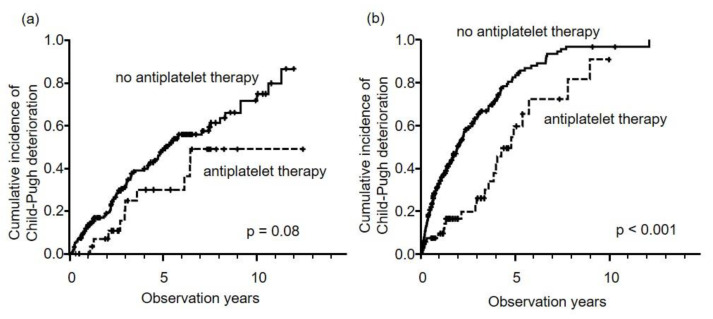
Cumulative incidence of Child–Pugh deterioration stratified by first treatment methods, i.e., curative therapy (**a**) and TACE (**b**). In curative therapy, five-year cumulative incidence rates of antiplatelet therapy (dotted line) and no antiplatelet therapy (solid line) were 30.0% and 48.6%, respectively (*p* = 0.08). In TACE (**b**), five-year cumulative incidence rates of antiplatelet therapy (dotted line) and no antiplatelet therapy (solid line) were 59.6% and 82.4%, respectively (*p* < 0.001).

**Figure 4 cancers-12-03215-f004:**
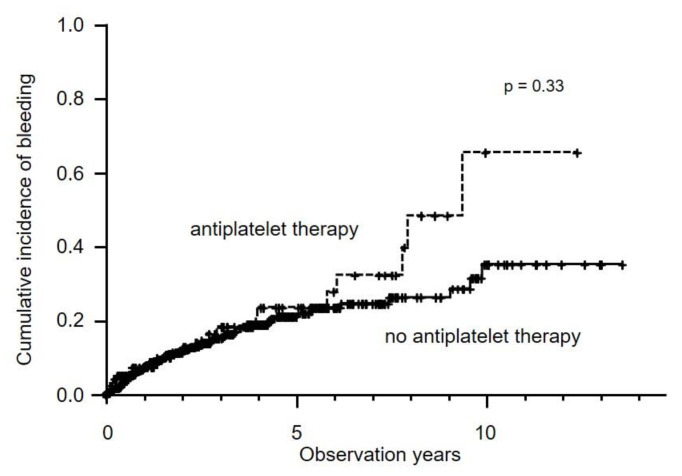
Cumulative incidence of hemorrhagic events in all patients. Five-year incidence rates for antiplatelet therapy (dotted line) and no antiplatelet therapy (solid line) were 23.1% and 21.4%, respectively (*p* = 0.33).

**Table 1 cancers-12-03215-t001:** Baseline characteristics in hepatocellular carcinoma (HCC) patients undertaking or not undertaking antiplatelet therapy.

Factor	Antiplatelet Therapy (*n* = 111)	No Antiplatelet Therapy (*n* = 661)	*p*-Value
Age (IQR), years	77 (69–82)	71 (64–78)	<0.01
Male, %	96 (86.5)	455 (68.9)	<0.01
Etiology, HBV/HCV/NBNC	14/46/51	84/345/231	0.13
Albumin (IQR), g/dL	3.9 (3.5–4.1)	3.6 (3.2–4.0)	<0.01
Bilirubin (IQR), mg/dL	0.6 (0.5–0.8)	0.8 (0.6–1.2)	<0.01
ALT (IQR), IU/L	29 (24–59)	36 (19–47)	<0.01
PLT (IQR), ×10^4^/μL	15.9 (11.4–19.0)	12.5 (8.7–17.8)	<0.01
Prothrombin time (IQR), %	85.2 (78.9–96.7)	79.3 (68.2–91.5)	<0.01
Child–Pugh grade (A/B/C)	97/11/3	450/176/35	<0.01
Stage, I/II/III/IVa/IVb	18/55/29/2/7	125/269/160/65/42	0.053
Tumor number (IQR)	1 (1–2)	1(1–3)	0.01
Tumor size (IQR), cm	3.3 (2.2–5.5)	2.8 (1.8–5.3)	0.05
Vascular invasion (%)	6 (5.4)	79 (12.0)	0.06
Extrahepatic metastasis (%)	7 (6.3)	43 (6.5)	1.0
AFP (IQR), ng/mL	8.8 (4.6–36.3)	15.3 (5.6–121.8)	0.01
DCP (IQR), mAU/mL	113.0 (29.5–951.5)	71.5 (25.0–845.0)	0.22
First treatment, curative/TACE/others/BSC	37/66/5/3	235/357/24/45	0.33

Continuous numeric variables are expressed as median and interquartile range (IQR). *p*-values are results that compare antiplatelet therapy with no antiplatelet therapy. Categorical variables were analyzed using the χ2-test or Fisher’s exact test, and continuous variables were compared using the Mann–Whitney U test. NBNC means non-hepatitis B virus (HBV) and non-hepatitis C virus (HCV). Abbreviations: AFP, alpha-fetoprotein; BSC, best supportive care; DCP, des-gamma-carboxy prothrombin; TACE, transcatheter arterial chemoembolization.

**Table 2 cancers-12-03215-t002:** Baseline factors associated with liver-related deaths in all patients in univariate and multivariate Cox progression analyses.

Factor	Univariate Analysis			Multivariate Analysis		
	HR	95% CI	*p*-Value	HR	95% CI	*p*-Value
Age, years						
≤70	1					
>70	1.065	0.873–1.30	0.533			
Gender						
Male	1					
Female	0.979	0.796–1.20	0.841			
Child–Pugh score	1.48	1.41–1.56	<0.001	1.37	1.29–1.47	<0.001
HCC stage						
I	1			1		
II	1.33	0.988–1.78	<0.001	1.20	0.847–1.71	0.303
III	2.75	2.03–3.73	<0.001	2.05	1.40–3.00	<0.001
IVa	13.8	9.45–20.0	<0.001	6.36	3.91–10.3	<0.001
IVb	21.3	14.1–32.1	<0.001	8.77	5.2–14.8	<0.001
AFP, ng/mL						
≤200	1			1		
>200	2.90	2.33–3.60	<0.001	1.36	1.05–1.77	0.019
DCP, mAU/mL						
≤400	1			1		
>400	3.01	2.47–3.66	<0.001	1.58	1.22–2.05	<0.001
First treatment						
Curative therapy	1			1		
TACE	3.15	2.46–4.01	<0.001	1.85	1.40–2.45	<0.001
Others	28.7	20.2–40.8	<0.001	4.24	2.70–6.66	<0.001
Antiplatelet therapy						
Without	1			1		
With	0.550	0.398–0.758	<0.001	0.640	0.445–0.926	0.002

Abbreviations: AFP, alpha-fetoprotein; DCP, des-gamma-carboxy prothrombin; TACE, transcatheter arterial chemoembolization.

**Table 3 cancers-12-03215-t003:** Baseline factors associated with tumor progression in patients who underwent TACE in univariate and multivariate Cox regression analyses.

Factor	Univariate Analysis			Multivariate Analysis		
	HR	95% CI	*p*-Value	HR	95% CI	*p*-Value
Age, years						
≤70	1					
>70	0.916	0.730–1.15	0.45			
Gender						
Male	1					
Female	1.22	0.943–1.57	0.13			
Child–Pugh score	1.04	0.957–1.13	0.352			
HCC stage						
I	1			1		
II	1.39	0.895–2.16	0.143	1.23	0.774–1.960	0.379
III	3.06	1.96–4.77	<0.001	2.47	1.52–4.01	<0.001
IVa	6.63	3.76–11.7	<0.001	5.59	3.00–10.4	<0.001
IVb	13.2	7.18–24.4	<0.001	8.83	4.47–17.4	<0.001
AFP, ng/mL						
≤200	1			1		
>200	1.63	1.26–2.11	<0.001	0.959	0.720–1.28	0.774
DCP, mAU/mL						
≤400	1			1		
>400	2.23	1.76–2.83	<0.001	1.63	1.25–2.12	<0.001
Antiplatelet therapy						
Without	1			1		
With	0.622	0.446–0.869	0.005	0.690	0.484–0.983	0.040

Abbreviations: AFP, alpha-fetoprotein; DCP, des-gamma-carboxy prothrombin; TACE, transcatheter arterial chemoembolization.

**Table 4 cancers-12-03215-t004:** Baseline factors associated with Child–Pugh deterioration in patients with Child–Pugh grade A liver disease at baseline in univariate and multivariate Cox regression analyses.

Factor	Univariate Analysis			Multivariate Analysis		
	HR	95% CI	*p*-Value	HR	95% CI	*p*-Value
Age, years						
≤70	1			1		
>70	1.32	1.04–1.67	0.023	1.69	1.29–2.20	<0.001
Gender						
Male	1					
Female	0.81	0.633–1.04	0.094			
Child–Pugh score						
5	1			1		
6	2.82	2.23–3.56	<0.001	3.46	2.69–4.46	<0.001
HCC stage						
I	1			1		
II	1.15	0.834–1.58	0.396	0.784	0.555–1.11	0.169
III	2.26	1.60–3.18	<0.001	1.36	0.925–1.99	0.119
IVa	12.1	7.56–19.3	<0.001	4.56	2.61–7.98	<0.001
IVb	13.3	7.78–22.8	<0.001	5.93	3.02–11.7	<0.001
AFP, ng/mL						
≤200	1			1		
>200	2.51	1.90–3.30	<0.001	1.80	1.31–2.47	<0.001
DCP, mAU/mL						
≤400	1			1		
>400	2.17	1.71–2.77	<0.001	1.55	1.17–2.06	<0.001
First treatment						
Curative therapy	1			1		
TACE	2.54	1.99–3.26	<0.001	1.77	1.34–2.36	<0.001
Others	24.3	13.9–42.4	<0.001	4.57	2.38–8.81	<0.001
Antiplatelet therapy						
Without	1			1		
With	0.524	0.369–0.746	<0.001	0.467	0.321–0.679	<0.001

Abbreviations: AFP, alpha-fetoprotein; DCP, des-gamma-carboxy prothrombin; TACE, transcatheter arterial chemoembolization.

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
