# Peer review of "Antiplatelet Therapy Improves the Prognosis of Patients with Hepatocellular Carcinoma"

_cancers, 2020, doi:10.3390/cancers12113215_

Round 1

Reviewer 1 Report

Interesting retrospective data and well designed study. 

  1. would recommend to clarify in the manuscript and in the introduction which data is related to aspirin specifically (rather to antiplatelet drug in general). 
  2. Background on the anti-cancer effect of clopidogrel in the introduction can be improved. 
  3. in the data reviewed, what % of pt had aspirin and what % had clopidogrel, might need some clarifications.  

Author Response

Response to Reviewer 1 Comments

Thank you for your kind review of our manuscript. Taking into all the questions and criticism raised by reviewers, we have now revised our manuscript. The responses are highlighted by red color text. The paper was submitted to MDPI English proofreading and proofread. We believe that we have successfully answered to all questions and our manuscript have improved to be suitable for publication. We look forward to hearing from you.

Point 1: would recommend to clarify in the manuscript and in the introduction which data is related to aspirin specifically (rather to antiplatelet drug in general). 

Response 1: About HCC incidence, aspirin showed lower risk of HCC than clopidogrel in previous study. We corrected it (line 44-46).

Point 2: Background on the anti-cancer effect of clopidogrel in the introduction can be improved. 

Response 2: We corrected it (line 51-54).

Point 3: in the data reviewed, what % of pt had aspirin and what % had clopidogrel, might need some clarifications.  

Response 3: The numbers of patients who took aspirin, clopidogrel and both were 96 (86.5%), 18(16.2%) and 3 (2.7%). We corrected it (line 72).

Reviewer 2 Report

This manuscript “Antiplatelet therapy improves the prognosis of patients with hepatocellular carcinoma” investigated the effects of antiplatelet therapy on prognosis, tumor progression, liver function and safety in HCC patients. Although this is not the first report showing the effect of antiplatelet therapy in HCC, the sample size is large

with various etiologies and observed for a maximum 13.6 years to make conclusions. This manuscript is well-written and has impact in the field particularly from the perspective point of therapeutic strategy. Some comments to strength this study are the following:

  1. Could the authors discuss the roles of platelet and thrombosis in HCC? The effect of antiplatelet therapy on HCC derives from anti-inflammation effect and anti-tumor effect. Are those effects related to platelet function?
  2. Is the effect of antiplatelet different in HCC patients with hepatitis B, hepatitis C, NASH, and cirrhosis?
  3. Could the authors include ALT, platelet, and MELD score in the baseline characteristics in HCC patients (Table 1).
  4. Do those HCC patients with antiplatelet therapy also take other medicines that can also improve HCC?

Author Response

Response to Reviewer 2 Comments

This manuscript “Antiplatelet therapy improves the prognosis of patients with hepatocellular carcinoma” investigated the effects of antiplatelet therapy on prognosis, tumor progression, liver function and safety in HCC patients. Although this is not the first report showing the effect of antiplatelet therapy in HCC, the sample size is large with various etiologies and observed for a maximum 13.6 years to make conclusions. This manuscript is well-written and has impact in the field particularly from the perspective point of therapeutic strategy. Some comments to strength this study are the following:

Thank you for your kind review of our manuscript. Taking into all the questions and criticism raised by reviewers, we have now revised our manuscript. The responses are highlighted by red color text. The paper was submitted to MDPI English proofreading and proofread. We believe that we have successfully answered to all questions and our manuscript have improved to be suitable for publication. We look forward to hearing from you.

Point 1: Could the authors discuss the roles of platelet and thrombosis in HCC? The effect of antiplatelet therapy on HCC derives from anti-inflammation effect and anti-tumor effect. Are those effects related to platelet function?

Response 1: Platelet activation causes tumor proliferation and metastasis. We added (line 234-239).

Point 2: Is the effect of antiplatelet different in HCC patients with hepatitis B, hepatitis C, NASH, and cirrhosis?

Response 2: According to the etiology, in NBNC the effect was significant. However, in HBV and HCV, the difference was not significant (p=0.07) We added it (line102-105)

Point 3: Could the authors include ALT, platelet, and MELD score in the baseline characteristics in HCC patients (Table 1).

Response 3: We added them (table1). However, it was difficult to show MELD score.

Point 4: Do those HCC patients with antiplatelet therapy also take other medicines that can also improve HCC?

Response 4: Medicines that have antitumor effect (such as metformin, NSAIDs and BCAA) are not discussed. However, in this study, the number of patients using these drugs from diagnosis of HCC to the endpoint was a few. Therefore, we could not take into consideration. This point was added in limitation (Line 255-258).

Reviewer 3 Report

Hayashi et al., evaluated the potential role of aspirin and clopidogrel in hepatocellular carcinoma (HCC) based on a retrospective analysis. While the manuscript is well written, I have a few comments to make it a better piece. 

1) Can authors clarify if they have performed propensity matched analysis between the anti-platelet agent cohort and the controls? If so, state so- this would improve the soundness of the manuscript. If this was not performed at the baseline, I think such an analysis (may be in supplement) should help in better analyzing the results.

If the propensity matched analysis is not performed- I am really worried about the credibility of the results- I see a wide difference at baseline characteristics of the patients in either cohorts- For instance, there is a difference in vascular invasion and AFP levels at the time of presentation. 

2) It is little unclear from the methods section when exactly the patients started taking anti-platelet agents? Also, did authors notice any dose effect and length of anti-platelet agents used in these patients? If such information is not available, it would be great if the same is listed in the limitations section. 

3) In line 82, Is MST a substitute for median OS? If yes, it would be advisable to use most widely accepted terminology. 

4) It would be great if authors can add a table detailing the inclusion and exclusion criteria.

5)Minor comment: It would be great if authors include p-values in the actual figures to make it easy for interpretation. 

Author Response

Response to Reviewer 3 Comments

Hayashi et al., evaluated the potential role of aspirin and clopidogrel in hepatocellular carcinoma (HCC) based on a retrospective analysis. While the manuscript is well written, I have a few comments to make it a better piece. 

Thank you for your kind review of our manuscript. Taking into all the questions and criticism raised by reviewers, we have now revised our manuscript. The responses are highlighted by red color text. The paper was submitted to MDPI English proofreading and proofread.We believe that we have successfully answered to all questions and our manuscript have improved to be suitable for publication. We look forward to hearing from you.

Point 1: 
Can authors clarify if they have performed propensity matched analysis between the anti-platelet agent cohort and the controls? If so, state so- this would improve the soundness of the manuscript. If this was not performed at the baseline, I think such an analysis (may be in supplement) should help in better analyzing the results.

If the propensity matched analysis is not performed- I am really worried about the credibility of the results- I see a wide difference at baseline characteristics of the patients in either cohorts- For instance, there is a difference in vascular invasion and AFP levels at the time of presentation.

Response 1: In propensity score matched analysis, antiplatelet therapy was associated with better prognosis. We added it (supplementary Table1 and Figure2).

Point 2: It is little unclear from the methods section when exactly the patients started taking anti-platelet agents? Also, did authors notice any dose effect and length of anti-platelet agents used in these patients? If such information is not available, it would be great if the same is listed in the limitations section. 

Response 2: We do not exactly know when patients started antiplatelet therapy. However, they received antiplatelet therapy at least in diagnosis of HCC. We added it in limitation (Line 258-260).

Point 3: In line 82, Is MST a substitute for median OS? If yes, it would be advisable to use most widely accepted terminology. 

Response 3: We changed MST to median OS (Line 89).

Point 4: It would be great if authors can add a table detailing the inclusion and exclusion criteria.

Response 4: We added it (Supplementary Figure1).

Point 5: Minor comment: It would be great if authors include p-values in the actual figures to make it easy for interpretation. 

Response 5:  We added it in all figures.

Round 2

Reviewer 3 Report

No comments